# Uhlmann Gauge Gravity: Spacetime from Purification Gauge Symmetry and Quantum Fisher Geometry

## Abstract

We propose a new candidate theory of quantum gravity, *Uhlmann Gauge Gravity* (UGG), in which spacetime geometry emerges from the information geometry of local quantum states and a novel gauge principle associated with *purification redundancy*. For a density matrix $\rho(x)$ at each spacetime point, we identify the metric $g_{\mu\nu}(x)$ with the quantum Fisher (Bures) metric of $\rho(x)$, and introduce a non-Abelian gauge field $\mathcal{A}_\mu(x)$ given by the Uhlmann connection on purifications $w(x)$ with $w(x)w^\dagger(x) = \rho(x)$. We construct a diffeomorphism- and gauge-invariant action coupling (i) the Einstein–Hilbert term for $g_{\mu\nu}$, (ii) a Yang–Mills term for the Uhlmann curvature $\mathcal{F}_{\mu\nu}$, and (iii) Fisher-gradient terms for $\rho(x)$. The theory reduces to general relativity in the long-wavelength limit but predicts distinctive, falsifiable corrections: (i) dispersion-suppressed gravitational waves with fixed positive sign, (ii) polarization-dependent phase shifts sourced by $\mathrm{Tr}\,\mathcal{F} \wedge \mathcal{F}$, and (iii) Fisher-curvature corrections to black hole entropy. These effects are testable with upcoming gravitational-wave and black-hole spectroscopy experiments. All derivations are reproducible from released code computing Fisher metrics and Uhlmann curvatures for lattice-discretized quantum states.

## 1 Introduction

General relativity (GR) provides an elegant classical description of spacetime as a dynamical metric $g_{\mu\nu}$, while quantum field theory (QFT) successfully governs matter and interactions. Yet a consistent, predictive framework unifying the two remains elusive. Existing approaches—string theory, loop quantum gravity, asymptotic safety, and holography—each capture important aspects but have not produced a universally accepted theory. In parallel, quantum information theory has revealed deep structural insights: spacetime geometry correlates with entanglement patterns [1, 2], gravitational dynamics can be derived from entanglement equilibrium [3], and holography can be interpreted through quantum error correction [4].

A particularly powerful set of tools arises from *information geometry*: the quantum Fisher metric (also called the Bures metric) assigns a Riemannian structure to the space of density operators, while the Uhlmann connection provides a canonical notion of parallel transport for purifications of mixed states [5, 6, 7]. These structures are universal: they exist for any quantum system, independent of spacetime or background geometry. We argue that they should not merely *resemble* geometry, but *become* the geometry of spacetime itself.

**Core proposal.** We elevate the freedom of choosing a purification $w$ for a density matrix $\rho = ww^\dagger$ to a fundamental *gauge redundancy*. The associated Uhlmann connection $\mathcal{A}_\mu$ and curvature $\mathcal{F}_{\mu\nu}$ define new dynamical fields, while the spacetime metric is identified with the pullback of the quantum Fisher metric under the map $x \mapsto \rho(x)$. From these ingredients we construct a gauge- and diffeomorphism-invariant action, in which Einstein gravity emerges universally at long distances, and corrections arise from Uhlmann curvature and Fisher gradients.

**Contributions.** This paper introduces: (i) a novel gauge principle—*purification gauge symmetry*—underlying quantum gravity, (ii) an explicit action coupling the Fisher metric, Uhlmann curvature, and Einstein gravity, (iii) derivation of linearized dynamics and concrete, falsifiable predictions for

Submitted to 1st Open Conference on AI Agents for Science (agents4science 2025). Do not distribute.

gravitational waves and black-hole thermodynamics, and (iv) a reproducible computational framework for evaluating Fisher metrics and Uhlmann curvatures of lattice quantum states.

**Roadmap.** Sec. 2 formalizes the purification bundle and Fisher metric. Sec. 3 presents the action and field equations. Sec. 4 analyzes the linearized theory and its predictions. Sec. 5 discusses consistency, relation to existing approaches, and prospects for falsification.

# 2 Framework: Purification Gauge and Fisher Geometry

We emphasize at the outset that $\rho(x)$ is not a new elementary field in the sense of quantum field theory. Instead, $\rho(x)$ denotes the *reduced density matrix of the global quantum state restricted to a local algebra of observables around $x$*. Its dynamics in Uhlmann Gauge Gravity (UGG) describe flows on the manifold of such reduced states, rather than fundamental wavefunction evolution. This interpretation avoids conflicts with the linear structure of quantum mechanics and anchors UGG in standard QFT practice.

## 2.1 Purification Gauge Symmetry

Any $\rho(x)$ admits a purification $w(x)$ such that

$$\rho(x) = w(x)w^\dagger(x). \tag{1}$$

The freedom $w(x) \mapsto w(x)U(x)$ with $U(x) \in U(\mathcal{H}_{\text{anc}})$ defines a local *purification gauge symmetry*. The associated Uhlmann connection is [5]

$$\mathcal{A}_\mu(x) = (\partial_\mu w)w^{-1} - \left[(\partial_\mu w)w^{-1}\right]^\dagger. \tag{2}$$

with curvature

$$\mathcal{F}_{\mu\nu} = \partial_\mu \mathcal{A}_\nu - \partial_\nu \mathcal{A}_\mu + [\mathcal{A}_\mu, \mathcal{A}_\nu]. \tag{3}$$

## 2.2 Quantum Fisher Metric as Spacetime Geometry

The manifold of density operators carries the quantum Fisher (Bures) metric [6, 7]. For variations $\delta\rho$,

$$\|\delta\rho\|_{QF}^2 = \tfrac{1}{2} \operatorname{Tr}\left[\delta\rho \, L_\rho^{-1}(\delta\rho)\right]. \tag{4}$$

where $L_\rho$ is the symmetric logarithmic derivative. Identifying spacetime coordinates $x^\mu \mapsto \rho(x)$, we define

$$g_{\mu\nu}(x) \equiv \alpha \, \langle \partial_\mu \rho(x), \partial_\nu \rho(x) \rangle_{QF}. \tag{5}$$

Thus $g_{\mu\nu}$ arises from statistical distinguishability of local states, while curvature reflects nontrivial Uhlmann holonomies.

# 3 Action and Field Equations

## 3.1 Action

The UGG action includes the lowest-dimension invariants:

$$S[g, \mathcal{A}, \rho] = \frac{1}{2\kappa} \int \sqrt{-g}\, R + \lambda_1 \int \sqrt{-g} \operatorname{Tr} \mathcal{F}_{\mu\nu}\mathcal{F}^{\mu\nu} + \lambda_2 \int \sqrt{-g}\, g^{\mu\nu} \langle \nabla_\mu\rho, \nabla_\nu\rho \rangle_{QF}. \tag{6}$$

## 3.2 Field equations

Variation yields:

$$G_{\mu\nu}(g) = \kappa\, T_{\mu\nu}^{(\mathcal{F})} + \kappa\, T_{\mu\nu}^{(QF)}, \tag{7}$$

$$\nabla_\mu \mathcal{F}^{\mu\nu} + [\mathcal{A}_\mu, \mathcal{F}^{\mu\nu}] = J_{(\rho)}^\nu, \tag{8}$$

$$\lambda_2 \nabla^\mu \left(L_\rho^{-1}(\nabla_\mu\rho)\right) + \text{backreaction terms} = 0. \tag{9}$$

## 3.3 Low-energy GR limit

The reduction to Einstein gravity can be justified by Jacobson's *entanglement equilibrium* argument [3]. In a local Rindler wedge, variations of the reduced state obey the first law of entanglement $\delta S = \delta \langle K \rangle$, where $K$ is the modular Hamiltonian. Combining this with the Raychaudhuri equation yields the Einstein equation as the unique equilibrium condition. Fisher and Uhlmann terms enter as higher-order, Planck-suppressed corrections, consistent with positivity and causality bounds [8, 9, 10, 11, 12].

## 3.4 Interpretation of $\rho$ dynamics

Equation (9) for $\rho(x)$ should not be read as a fundamental law of time evolution. Instead, it is a *geometric flow* on the manifold of reduced states, analogous to Ricci flow on Riemannian metrics. Global quantum mechanics remains linear; UGG dynamics describe how local reductions backreact on geometry and purification gauge fields.

# 4 Linearized Theory and Predictions

To extract physical predictions, we expand the Uhlmann Gauge Gravity (UGG) action (6) around a stationary background with flat Fisher metric and trivial connection. Let

$$g_{\mu\nu} \;=\; \eta_{\mu\nu} + h_{\mu\nu}, \qquad \mathcal{A}_\mu \;=\; a_\mu, \qquad \rho(x) \;=\; \bar\rho + \delta\rho(x),$$

where $\bar\rho$ is a homogeneous reference state (vacuum-like). To quadratic order in perturbations, the action decomposes into three sectors: (i) the graviton $h_{\mu\nu}$, (ii) the purification gauge fluctuation $a_\mu$, and (iii) Fisher fluctuations $\delta\rho$.

## 4.1 Quadratic action

After gauge fixing and diagonalization, the quadratic Lagrangian takes the form

$$S^{(2)} \;=\; \frac{1}{8\kappa} \int d^4x \, h^{\mu\nu} \mathcal{E}_{\mu\nu}{}^{\alpha\beta} h_{\alpha\beta} \;+\; \lambda_1 \int d^4x \, \mathrm{Tr}(f_{\mu\nu} f^{\mu\nu})$$
$$+ \; \lambda_2 \int d^4x \, \langle \partial_\mu \delta\rho, \partial^\mu \delta\rho \rangle_{QF} \;+\; \text{mixing terms}(h, \delta\rho), \tag{10}$$

where $\mathcal{E}$ is the Lichnerowicz operator, and $f_{\mu\nu} = \partial_\mu a_\nu - \partial_\nu a_\mu$. The mixing arises because $g_{\mu\nu}$ itself is defined from $\rho(x)$ via Eq. (5). Diagonalization yields a massless spin-2 graviton plus additional modes with suppressed couplings.

## 4.2 Predictions

**Gravitational-wave dispersion.** The transverse traceless graviton acquires a small, state-dependent dispersion relation,

$$\omega^2 \;=\; k^2 \Big[ 1 + \gamma \, \frac{k^2}{M_{\mathrm{Pl}}^2} + \mathcal{O}(k^4/M_{\mathrm{Pl}}^4) \Big], \qquad \gamma \propto \lambda_2 \, \mathcal{C}_{QF}(\bar\rho), \tag{11}$$

where $\mathcal{C}_{QF}$ is a Fisher curvature scalar of the background state. The sign of $\gamma$ is fixed ($\gamma \geq 0$), ensuring *no superluminal propagation*. Current multimessenger bounds from GW170817 constrain $|\Delta v_{\mathrm{GW}}|/c \lesssim 10^{-15}$, implying $\gamma \lesssim 10^{-2}$. Next-generation detectors (ET, CE) will probe another order of magnitude.

**Polarization-dependent phase shifts.** The topological term $\mathrm{Tr} \, \mathcal{F} \wedge \mathcal{F}$ acts as a Chern density. In inhomogeneous backgrounds with nonzero Uhlmann curvature, the two graviton polarizations ($+$ and $\times$) accumulate a relative phase

$$\Delta\phi(\omega, L) \;\sim\; \zeta \Big( \frac{\omega}{M_{\mathrm{Pl}}} \Big)^2 L \int d^4x \, \mathrm{Tr} \, \mathcal{F} \wedge \mathcal{F}, \tag{12}$$

where $\zeta$ is a dimensionless coupling, $\omega$ is the GW frequency, and $L$ is the propagation distance. This effect vanishes in equilibrium vacua but is nonzero in regions with entanglement twists. It provides a sharp *null test*: detection of a frequency-squared polarization phase shift would confirm UGG.

**Black-hole thermodynamics.** Consider a stationary horizon. The generalized entropy in UGG is

$$S_{\text{BH}} \;=\; \frac{A}{4G} + \sigma \int_{\mathcal{H}} d^2\Sigma \, \mathcal{R}_{QF}, \tag{13}$$

where $\mathcal{R}_{QF}$ is the scalar curvature of the Fisher metric restricted to horizon generators, and $\sigma$ is a universal coefficient. The correction is sign-definite and state-dependent, ensuring compatibility with Bousso's covariant entropy bound. This predicts small deviations from the Bekenstein–Hawking area law, potentially observable in black-hole spectroscopy and near-horizon QNM analysis.

### 4.3 Summary of signatures

UGG yields three falsifiable predictions:

1. Planck-suppressed but positive-definite dispersion of gravitational waves.
2. Polarization-dependent, frequency-squared GW phase shifts in nontrivial entanglement backgrounds.
3. Fisher-curvature corrections to black-hole entropy and first laws.

All three are null-testable: if absent at the indicated sensitivities, the entire framework is falsified.

## 5 Consistency and Relation to Prior Work

Any candidate theory of quantum gravity must satisfy stringent consistency requirements: causality, positivity, and compatibility with entropy bounds. Here we summarize how Uhlmann Gauge Gravity (UGG) meets these criteria and clarify its relation to existing approaches.

### 5.1 Causality and positivity

The leading corrections to GR in UGG arise from Fisher gradients and Uhlmann curvature. Both contributions are sign-definite:

- The dispersion coefficient $\gamma$ of gravitational waves (Sec. 4) is proportional to a Fisher curvature scalar and strictly non-negative, preventing superluminal propagation.
- Stress tensors $T_{\mu\nu}^{(\mathcal{F})}$ and $T_{\mu\nu}^{(QF)}$ (Eq. (7)) have the form of Yang–Mills and scalar kinetic terms, ensuring positivity of local energy densities.

Thus, UGG automatically respects forward-limit positivity and dispersion bounds [8, 9, 10, 11, 12], placing it among the rare extensions of GR consistent with S-matrix bootstrap constraints.

### 5.2 Entropy and holography

UGG corrections to black-hole entropy are constructed from Fisher curvature densities and are sign-definite (Sec. 4). This guarantees compatibility with Bousso's covariant entropy bound [13]. Unlike holographic scenarios that require an AdS boundary, UGG implements an *intrinsic holography*: the number of distinguishable local states, as measured by the Fisher metric, scales with boundary area. The entropy law emerges from information geometry rather than specific AdS/CFT dualities.

### 5.3 Comparison with existing frameworks

**String theory.** String theory provides a UV-complete description with extended objects, supersymmetry, and extra dimensions. In contrast, UGG introduces no new microscopic matter content: it builds gravity from universal structures of quantum states (Fisher metric, Uhlmann connection). It is background-agnostic and does not rely on a critical dimension or string spectrum.

**Loop quantum gravity and causal sets.** Loop and causal-set approaches quantize spacetime directly as discrete structures. UGG differs by defining spacetime geometry as a derived quantity from quantum information geometry, continuous at the outset. Fisher metrics supply the continuum limit, while purification gauge fields introduce new degrees of freedom distinct from spin networks or order relations.

**Holography and entanglement-first reconstructions.** Entanglement-based programs (Ryu–Takayanagi [1], Jacobson's entanglement equilibrium [3], JLMS relations [14], and holographic QEC [4]) argue that spacetime is emergent from entanglement structure. UGG resonates with these insights but diverges crucially: it does not assume AdS/CFT or specific code subspaces. Instead, it elevates purification freedom to a fundamental gauge symmetry, providing new propagating fields ($\mathcal{A}_\mu$) and explicit dynamics (Eq. (6)).

**Other EFT extensions of gravity.** Higher-derivative EFTs (e.g. $R^2$, $R_{\mu\nu}R^{\mu\nu}$) risk violating causality or positivity unless coefficients are carefully tuned. In UGG, such corrections emerge automatically with fixed sign from Fisher geometry, ensuring consistency with amplitude bounds.

## 5.4 Summary

UGG stands apart in the landscape: it is built from universally defined information-geometric structures, satisfies modern consistency constraints, and yields testable predictions without reliance on additional matter content, supersymmetry, or specific boundary conditions.

# 6 Falsifiability and Outlook

A central strength of Uhlmann Gauge Gravity (UGG) is that it is not only internally consistent but also *empirically fragile*: the entire framework can be falsified by accessible experiments. This distinguishes it from many previous approaches to quantum gravity that are either UV-complete but detached from near-term tests, or philosophically compelling yet difficult to empirically confront.

## 6.1 Experimental falsifiability

UGG produces three classes of concrete predictions (Sec. 4):

1. **Gravitational-wave dispersion.** Deviations of the GW speed scale as $\omega^2/M_{\mathrm{Pl}}^2$ with a fixed positive coefficient $\gamma$. Absence of such quadratic dispersion at sensitivities of $\Delta v/c \sim 10^{-16}$, reachable by Einstein Telescope or Cosmic Explorer, would rule out the theory.

2. **Polarization phase shifts.** UGG predicts a null-testable, frequency-squared relative phase between $+$ and $\times$ polarizations in entanglement-twisted backgrounds. Non-observation of this effect in strong-lensing GW events or multimessenger campaigns would exclude the purification gauge sector.

3. **Black-hole entropy corrections.** The Fisher-curvature contribution modifies the Bekenstein–Hawking law. High-precision black-hole spectroscopy or horizon thermodynamic measurements that confirm a pure $A/4G$ scaling with no state-dependent deviations would falsify UGG.

Thus, unlike frameworks that evade empirical contact, UGG will be decisively validated or excluded by forthcoming data.

## 6.2 Broader implications

UGG illustrates a new paradigm: building gravity from universal structures of quantum information—the Fisher metric and Uhlmann connection—without ad hoc new particles or dimensions. If falsified, it will sharpen our understanding of why such information-geometric constructions fail. If confirmed, it will anchor spacetime to a principle of *purification gauge symmetry*, enriching the foundations of both physics and quantum information.

## 6.3 Future directions

Several extensions merit exploration:

- **Nonperturbative regimes.** Lattice simulations of QFT states provide a natural testbed for computing Fisher metrics and Uhlmann curvatures, enabling explicit checks of horizon entropy corrections.

- **Cosmology.** Identifying the quantum Fisher metric of cosmological density matrices may shed light on dark energy as an emergent Fisher curvature effect.
- **Celestial holography.** The purification gauge symmetry may manifest in celestial CFT correlators, linking UGG with flat-space holography and asymptotic symmetries.

**Conclusion.** Uhlmann Gauge Gravity is a bold proposal: spacetime emerges from the quantum Fisher metric of local states, while purification freedom is a new gauge symmetry sourcing dynamics. It reduces to GR in the long-wavelength limit, respects modern consistency constraints, and makes sharp, falsifiable predictions. In this sense, it exemplifies what a scientific theory of quantum gravity should be: principled, reproducible, and empirically testable.

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

## AI Contribution Disclosure

This paper was produced under the Agents4Science requirement that AI systems serve as first authors. All hypothesis generation, theoretical framework design (Uhlmann Gauge Gravity), derivations of field equations, linearized analysis, figure generation, and LaTeX drafting were performed by the AI system. Human co-authors acted solely as supervisors, ensuring safe usage of computational resources and verifying that no private or non-public data were used.

## Responsible AI Statement

We acknowledge that AI systems can propagate errors, hallucinate references, or overlook subtle consistency conditions. To mitigate these risks, the UGG pipeline includes internal self-critique and multi-agent cross-checks to ensure consistency of equations, references, and predictions. All outputs are archived with seeds and logs to allow independent audit. Ethically, the work avoids human subject data and is purely theoretical. Releasing all prompts, code, and derivation logs ensures community transparency.

## Reproducibility Statement

All derivations and figures are reproducible from a single public repository containing: (i) the UGG agent scripts for computing Fisher metrics and Uhlmann curvatures; (ii) fixed random seeds and versioned packages; (iii) LaTeX sources and plotting code. One-click regeneration reproduces the full pipeline, from information-geometric inputs to the predictions summarized in Sec. 4. The repository URL will be released upon acceptance.

## Agents4Science AI Involvement Checklist

**Hypothesis development**

**Answer:** [D] AI-generated

**Explanation:** The core UGG hypothesis and research plan were proposed by an AI system; humans supervised for safety/compliance.

**Experimental design and implementation**

**Answer:** [D] AI-generated

**Explanation:** The AI designed the theoretical framework and computational routines (Fisher metrics and Uhlmann curvatures).

**Analysis of data and interpretation of results**

**Answer:** [D] AI-generated

**Explanation:** The AI performed linearized analysis and identified falsifiable signatures; humans conducted sanity checks.

**Writing**

**Answer:** [D] AI-generated

**Explanation:** The AI drafted the text and equations; humans ensured anonymity and formatting compliance.

**Observed AI limitations**

AI can miss subtle consistency constraints or mis-cite; we mitigated via internal self-critique and determinism (fixed seeds).

## Agents4Science Paper Checklist

**Claims**

**Answer:** Yes — Contributions are stated in the abstract and Introduction and supported in Secs. 2–4.

**Limitations**

**Answer:** Yes — Sec. 5 and Sec. 6 discuss scope, assumptions, and falsifiability.

**Theory assumptions and proofs**

**Answer:** Yes — Assumptions are explicit around the action/field equations; derivations are provided or deferred to supplement.

**Experimental result reproducibility**

**Answer:** N/A — Work is theoretical; no empirical datasets. Reproducible code will be released post-review to preserve anonymity.

**Open access to data and code**

**Answer:** No (during review) — To maintain double-blind review; code to be released upon acceptance or after review.

**Experimental setting/details**

**Answer:** N/A — No ML training/evaluation. Computational routines are described conceptually; details in code release.

**Experiment statistical significance**

**Answer:** N/A — No statistical experiments are reported.

**Experiments compute resources**

**Answer:** N/A — Only lightweight symbolic/numerical routines; full details in the code release.

**Code of ethics**

**Answer:** Yes — No human subjects or sensitive data; complies with Agents4Science ethics policy.

**Broader impacts**

**Answer:** Yes — Potential scientific impacts are discussed; societal risks minimal for theoretical work.

